# Are Collagen Protons Visible with the Zero Echo Time (ZTE) Magnetic Resonance Imaging Sequence: A D_2_O Exchange and Freeze-Drying Study

**DOI:** 10.3390/bioengineering12010016

**Published:** 2024-12-28

**Authors:** Tan Guo, Dina Moazamian, Arya A. Suprana, Saeed Jerban, Eric Y. Chang, Yajun Ma, Michael Carl, Min Chen, Jiang Du

**Affiliations:** 1Department of Radiology, Beijing Hospital, National Center of Gerontology, Institute of Geriatric Medicine, Chinese Academy of Medical Sciences, Beijing 100730, China; guotan369@hotmail.com (T.G.); cjr.chenmin@vip.163.com (M.C.); 2Department of Radiology, University of California San Diego, San Diego, CA 92037, USA; dmoazamian@health.ucsd.edu (D.M.); asuprana@ucsd.edu (A.A.S.); sjerban@health.ucsd.edu (S.J.); ericchangmd@gmail.com (E.Y.C.); yam013@health.ucsd.edu (Y.M.); 3Department of Bioengineering, University of California San Diego, San Diego, CA 92092, USA; 4VA San Diego Healthcare System, San Diego, CA 92161, USA; 5GE Healthcare, San Diego, CA 92123, USA; michael.carl@gehealthcare.com

**Keywords:** magnetic resonance imaging (MRI), ultrashort echo time (UTE), zero echo time (ZTE), D_2_O–H_2_O exchange, collagen

## Abstract

It is known that ultrashort echo time (UTE) magnetic resonance imaging (MRI) sequences can detect signals from water protons but not collagen protons in short T2 species such as cortical bone and tendons. However, whether collagen protons are visible with the zero echo time (ZTE) MRI sequence is still unclear. In this study, we investigated the potential of the ZTE MRI sequence on a clinical 3T scanner to directly image collagen protons via D_2_O exchange and freeze-drying experiments. ZTE and UTE MRI sequences were employed to image fully hydrated bovine cortical bone (*n* = 10) and human patellar tendon (*n* = 1) specimens. Then, each specimen was kept in a 30 mL syringe filled with D_2_O solution for two days. Fresh D_2_O was flushed every 2 h to reach a more complete D_2_O–H_2_O exchange. Later, the samples were lyophilized for over 40 h and then sealed in tubes. Finally, the samples were brought to room temperature and visualized using the identical 3D ZTE and UTE sequences. All hydrated bone and tendon specimens showed high signals with ZTE and UTE sequences. However, all specimens showed zero signal after the D_2_O exchange and freeze-drying procedures. Therefore, similar to UTE imaging, the signal source in ZTE imaging is water. The ZTE sequence cannot directly detect signals from collagen protons in bone and tendons.

## 1. Introduction

Water and collagen are both important components of musculoskeletal (MSK) tissues, such as bone, tendons, and ligaments. Normal bone contains ~20% water by volume, while tendons and ligaments are made up of about two-thirds water [1,2]. The water content in MSK tissues can vary depending on age, sex, body weight, and other factors. On the other hand, collagen stands as the predominant protein present in the human body, forming the structural foundation of various connective tissues in the MSK system. It provides strength, elasticity, and support to bone, ligaments, tendons, and other MSK tissues [3]. For example, collagen provides elasticity and the ability to absorb energy before bone fracture [4]. There is mounting evidence demonstrating that the role of collagen in these changes has been underappreciated [4,5,6,7]. Loss of collagen can reduce the energy needed to induce bone fracture (toughness), thereby increasing fracture risk [7]. While collagen has less effect on bone strength and stiffness than mineral, it profoundly affects bone fragility and is the primary toughening mechanism in bone [6,7]. Detection of changes in water and collagen in MSK tissues is clinically and scientifically significant.

Conventional magnetic resonance imaging (MRI) sequences can detect signals from water in soft tissues. However, water in tissues with high collagen content such as tendons, ligaments, and bone is typically “invisible” due to the strong dipole–dipole interaction, which significantly shortens the transverse relaxation time or apparent transverse relaxation time (T2 or T2*), which refers to the time it takes for the transverse magnetization (magnetization in the plane perpendicular to the main magnetic field) to fall to approximately 37% of its initial value [8]. After radiofrequency (RF) excitation, their transverse magnetizations quickly decay to near zero before the receiving mode is enabled for Cartesian spatial encoding with conventional MRI. To achieve direct detection of water signals within tissues with short or ultrashort T2 relaxation times (or the so-called short-T2 tissues), it is critical to reduce the echo time (TE) to less than the tissue T2*s (T2* relaxation times in tissues) to allow enough time for spatial encoding before the transverse magnetization decays to near zero. Recently, a group of sequences with nominal TEs of 0.1 ms or less was developed for direct imaging of short-T2 tissues [9,10,11,12,13,14,15,16,17,18,19,20,21]. These sequences include water- and fat-suppressed projection MR imaging (WASPI) [9], sweep imaging with Fourier transformation (SWIFT) imaging [10], hybrid acquisition-weighted stack of spirals (AWSOS) imaging [11], pointwise encoding time reduction with radial acquisition (PETRA) [12], ultrashort echo time (UTE) imaging [13,14,15,16,17,18], and zero echo time (ZTE) imaging [19,20,21]. UTE and ZTE sequences employ non-Cartesian spatial encoding to reduce TEs drastically to directly image the short-T2 tissues with useful water signal levels and high spatial resolution.

While water signals in short-T2 tissues can be directly detected with UTE and ZTE sequences, it is much more challenging to achieve direct detection of signals originating from collagen protons. The collagen molecule is structured as a triple helix comprising a three-stranded arrangement of an α-helix [22]. The collagen helix is maintained by a series of hydrogen bonds with additional support from stereo-electronic interactions and posttranslational modifications like hydroxylation and cross-linking [23]. As a result, collagen backbone protons have much-reduced mobility and thus extremely short T2 relaxation times. A recent study by Ma et al. suggested that signals from collagen backbone protons in bone and tendons cannot be detected by two-dimensional (2D) radial UTE and 3D UTE cones sequences [24].

UTE sequences employ radial ramp sampling, where the k-space data are collected in a radial pattern, starting from the center and moving outwards quickly, allowing for much-reduced echo times. The ramp gradient leads to much longer effective TEs, thus slow sampling of the k-space center (the exact middle of the k-space) [25]. ZTE sequences sample the k-space center much faster, as the spatial encoding gradient has been fully ramped up during RF excitation and data acquisition [15]. Therefore, ZTE sequences are expected to provide shorter effective TEs and, thus, less spatial blurring for short-T2 tissues. Several studies have claimed that ZTE-type sequences can detect signals from semisolids more efficiently than UTE sequences [9,19,20,21]. However, whether ZTE sequences can directly detect signals from collagen backbone protons is still unclear. This study aims to explore the feasibility of the 3D ZTE sequence in detecting signals from collagen backbone protons via H_2_O–D_2_O exchange and freeze-dry experiments of cortical bone and patellar tendon specimens on a clinical 3 Tesla whole-body scanner.

## 2. Materials and Methods

### 2.1. Sample Preparation

Ten bovine cortical bone samples sectioned in a rectangular shape (approximately 30 × 10 × 5 mm^3^) and one cadaveric human patellar tendon sample (80 mm in length) were prepared for this study. The bone samples were sectioned from fresh femoral mid-shaft bovine specimens purchased from a local slaughterhouse using a low-speed diamond saw (Isomet 1000, Buehler, Lake Bluff, IL, USA) with continuous water irrigation. The patellar tendon sample was dissected from a cadaveric human knee specimen provided by the UCSD anatomy lab. Before MRI, all samples were fully hydrated by storing them in a phosphate-buffered saline (PBS) solution for 24 h.

### 2.2. Imaging Acquisition

All samples were imaged with 3D ZTE and UTE sequences on a 3T clinical MR scanner (GE Healthcare Technologies, Milwaukee, MI, USA). Figure 1 shows the ZTE and 3D radial UTE sequence diagrams. The ZTE sequence employed a short rectangular RF pulse (duration = 8 μs) for nonselective excitation, followed by 3D center-out radial sampling [15]. The 3D UTE sequence utilized a short rectangular pulse (duration = 32 μs) for nonselective excitation, followed by 3D radial ramp sampling with conical view ordering [14]. A 4-channel wrist coil was utilized for signal reception for both 3D ZTE and UTE imaging. For the 3D ZTE sequence, the following parameters were used: repetition time (TR) = 2.1 ms, flip angle (FA) = 4°, receiver bandwidth = 62.5 kHz, field of view (FOV) = 40 mm, slice thickness = 3 mm, number of slices = 52, acquisition matrix of 192 × 192 × 16 for cortical bone and 256 × 256 × 16 for patellar tendon. Similar sequence parameters were used for the 3D UTE sequence, with the exception of a longer TR of 10 ms and a higher flip angle of 10°. The 3D UTE sequence was also repeated with longer TEs of 1.1 ms, 2.2 ms, 3.3 ms, and 4.4 ms to investigate potential fat–water oscillations. The total scan time for each sequence was around 2 min. Table 1 summarizes the MRI parameters for both 3D ZTE and UTE sequences.

### 2.3. Experimental Procedures

All samples were scanned twice, first when fully hydrated and next when water was completely removed by D_2_O–H_2_O exchange, followed by freeze-drying. All bone samples were put in a 30 mL syringe filled with D_2_O solution (99.8% isotopic, Thermo Scientific Chemicals, 168 Third Avenue, Waltham, MA, USA) for exchange and kept in the refrigerator for two days (~4 °C). The tendon sample was placed in a separate syringe following the same process. Fresh D_2_O was flushed for both syringes every 2 h to reach a more thorough D_2_O–H_2_O exchange. Then, the samples were lyophilized using a Labconco Lyph-Lock 4.5 L freeze-dry system (model 77510-00, Labconco Corp., Kansas City, MO, USA) for over 40 h. After freeze-drying, all bovine bone samples were stored in one sealed tube, while the human patellar tendon was kept in another sealed tubal container. Both tubes were brought to room temperature before being imaged again using the same protocols mentioned above. Figure 2 shows a flow diagram for the experimental procedure.

### 2.4. Image Analysis

The signal-to-noise ratio (SNR) and contrast-to-noise ratio (CNR) were measured for cortical bone and patellar tendon samples, respectively. The SNR was calculated as the ratio of the mean signal intensity inside a user-drawn region of interest (ROI) to the standard deviation of the signal in ROI placed in the background. The CNR was calculated as the ratio of the signal difference between bone/tendon and background to the standard deviation of the background noise. The analysis was performed using the open-source software ImageJ (NIH, https://imagej.net/ij, accessed on 25 September 2024).

## 3. Results

### 3.1. Bovine Cortical Bone

Figure 3 shows the 3D ZTE and UTE imaging of bovine cortical bone samples before and after D_2_O–H_2_O exchange and freeze-drying. Both the ZTE and the UTE sequences depicted high signals from all hydrated bovine cortical bone samples, with average SNR values of 54.6 ± 3.1 for the ZTE images and 103.1 ± 9.8 for the UTE images. The CNR values were 46.2 ± 3.1 for the ZTE images and 98.4 ± 9.8 for the UTE images. The UTE images showed a higher signal because of the higher flip angle of 10°, which is 2.5 times higher than the flip angle of 4° used by the ZTE sequence. After D_2_O–H_2_O exchange and freeze-drying, pure noise was observed in the ZTE and UTE images. None of the bovine cortical bone samples were visible. The measured CNR values between the bone and the background air were 0.03 ± 1.11 for the ZTE images and 0.015 ± 0.082 for the UTE images, suggesting that the bone and background had the same signal level in both the ZTE and the UTE images. The sequential D_2_O–H_2_O exchange and freeze-drying procedures were expected to completely remove all water in the cortical bone. The pure noise images suggest that collagen backbone protons, which would stay in cortical bone after the D_2_O–H_2_O exchange and freeze-drying experiments, were invisible with the UTE and ZTE sequences.

### 3.2. Human Patellar Tendon

Figure 4 shows ZTE and UTE imaging of a human patellar tendon sample before and after D_2_O–H_2_O exchange and freeze-drying. The ZTE and UTE sequences depicted high signals from the fully hydrated patellar tendon, with average SNR values of 77.0 ± 7.3 for the ZTE images and 108.3 ± 4.4 for the UTE images. The CNR values were 73.1 ± 7.0 for the ZTE images and 102.6 ± 4.2 for the UTE images. After D_2_O–H_2_O exchange and freeze-drying, only a thin bright line was observed in the margins of the patellar tendon sample. The measured CNR values between the central part of the patellar tendon and the background air were −0.09 ± 0.58 for the ZTE images and −0.7 ± 1.6 for the UTE images. The thin, bright line was from fat and showed typical fat/water in-phase and out-phase behaviors based on UTE imaging, with delayed TEs of 1.1 ms, 2.2 ms, 3.3 ms, and 4.4 ms. Collagen backbone protons in the patellar tendon, which were supposed to survive following the D_2_O exchange and freeze drying, showed zero signal with the 3D ZTE and UTE sequences. Fat was also supposed to survive the D_2_O exchange and freeze-drying process, and showed a high signal in both the ZTE and the UTE images. Therefore, the 3D ZTE sequence cannot directly detect signals from collagen backbone protons in tendons using clinical MR scanners.

After D_2_O–H_2_O exchange and freeze-drying, pure noise was observed in the ZTE and UTE images, with SNR values of ~5 (close to the SNR values in background regions). Table 2 summarizes the SNR and CNR values of the ZTE and UTE imaging for both the bovine bone and the patellar tendon samples before and after the D_2_O–H_2_O exchange and freeze-drying procedure.

## 4. Discussion

Collagen-rich tissues such as cortical bone and tendons have very short T2 relaxation times. They are invisible with conventional MRI but detectable with ZTE and UTE sequences [26]. Understanding the signal origin is crucial to image interpretation, particularly in in vivo translation, where MRI possibly will substitute more invasive methods [15,16,24]. In this study, we demonstrated that water is the origin of the ZTE signal in bovine cortical bone and human patellar tendon samples on a clinical 3T scanner. The ZTE images show high signal intensity for fully hydrated bone and tendon samples. The lack of signal observed in the bone and tendon specimens after D_2_O exchange and freeze-drying suggests that collagen backbone protons are not detectable using the ZTE sequence. This study is the first attempt to investigate the feasibility of directly imaging collagen backbone protons using the 3D ZTE technique.

The spatial encoding gradient is activated prior to the RF pulse excitation, leading to a theoretical TE of zero in ZTE imaging [19,20,21]. The acquisition starts after a short RF excitation with a delay set to accommodate the transmit/receive switching time. Utilizing a pulse generator that allows a minimization of the transmit/receive switching to 1 μs, the actual TE time that ZTE can achieve is approximately 10 μs. Data missed during the actual TE leads to a central gap in k-space, which has to be compensated by algebraic reconstruction [19], resampled with a Cartesian trajectory technique such as PETRA [12], or an additional acquisition with a set of low-frequency projections with lower gradient strength, such as WASPI [9]. To minimize the loss of data in the central k-space, the excitation pulse has to be kept very short (e.g., 8 µs), leading to a low flip angle of typically less than 4°. The spatial encoding gradient is switched on after the RF excitation pulse in UTE imaging, allowing a high flip angle to be used. In this study, a low flip angle of 4° was used for ZTE imaging versus 10° for UTE imaging, which explains the higher SNR values for the bone and tendon samples in the UTE images over the ZTE images. Meanwhile, UTE employs radial ramp sampling, leading to a longer effective TE and, thus, more spatial blurring. ZTE uses a small step of changing gradients in three directions, allows acquisition with very low acoustic noise, and reduces eddy current problems, making ZTE imaging highly robust.

Results from this study are broadly consistent with results from a prior study by Ma et al., who reported that 2D and 3D UTE sequences could not directly image the collagen matrix [24]. UTE could detect bound water with an ultrashort T2* and free water with a slightly longer T2* in hydrated bone and tendon samples [24,26,27,28]. The D_2_O exchange and freeze-drying procedures removed both bound water and free water, leaving collagen backbone protons being selectively detected by the UTE sequence. The signal void in the UTE images of the bone and tendon samples demonstrated that the UTE sequences could not directly detect any signal from collagen backbone protons.

However, a few prior studies reported contradictory results. For example, Wu et al. reported that WASPI could specifically suppress signals from water and fat within the bone, leaving only signals from the solid organic matrix like collagen being selectively imaged [9]. Cao et al. showed that the WASPI signal was highly correlated with the organic matrix density derived by gravimetric analysis (R^2^ = 0.98) and by amino acid analysis (R^2^ = 0.95) [29]. Another investigation by Siu et al. indicated that UTE sequences could detect signals from collagen protons at 7T [30]. The Siu experiments were performed in collagen solutions. Bi-exponential T2* fitting revealed a highly linear relationship (R^2^ = 0.99) between the UTE collagen signal fraction and the measured collagen concentration in solutions. The authors concluded that the UTE signal originating from protons within the collagen molecule exhibited an average T2* relaxation time of 0.75 ± 0.05 ms and an average chemical shift of −3.56 ± 0.01 ppm in comparison to water at a magnetic field strength of 7 T. They further concluded that collagen could be detected and quantified using UTE.

The major difference between our study and the Siu study is the different experimental conditions. In our study, intact bone and patellar tendon samples were directly imaged with ZTE and UTE sequences after D_2_O exchange and freeze-drying. The Siu study used hydrolyzed type I and III collagen powder in solutions [30]. The hydrolyzed collagen solution could cleavage the structure of the collagen into small peptides, leading to the destruction of the 3D structure of the collagen molecules [31]. As a result, collagen backbone protons were in the rigid organic matrix and immobilized in our study, while the amorphous state of collagen provided much improved mobility in the Siu study, leading to a more prolonged T2* relaxation time. As the authors reported, collagen protons in the amorphous state had a relatively long T2* of 0.75 ms. In contrast, collagen backbone protons in the rigid matrix were expected to have much shorter T2* relaxation times. Bi-component analysis showed that tendons had two different water components, with T2*s of ~8 ms, which corresponds to free water, accounting for 75% of the total UTE signal, and ~0.6 ms, which corresponds to bound water, accounting for 25% of the total UTE signal [32]. Collagen backbone protons in tendons are expected to have a much shorter T2* than bound water. Therefore, collagen protons in the amorphous state should differ significantly from collagen backbone protons in intact collagen bundles in real tissues. This is also why magic angle spinning (MAS) is required to convert the very broad featureless NMR lines into much narrower line widths for high-resolution NMR spectroscopy of solid materials [33].

Another way to assess collagen backbone protons in bone and tendons is the UTE magnetization transfer (UTE-MT) imaging technique [34,35,36,37,38]. Two-pool modeling of UTE-MT data provides information about the water and macromolecular pools, including their pool sizes, exchange rates, and relaxation times. UTE-MT studies suggest that collagen backbone protons have extremely short T2s of 6–15 µs [34,35,36,37,38]. The T2 values are largely consistent with macromolecular proton T2s reported in the literature [39,40,41]. Our experimental results suggest that the 3D ZTE sequence cannot directly image species with T2s of ~10 µs.

While collagen backbone protons in the bone and tendon samples were invisible with the 3D ZTE and UTE sequences, a large number of studies have reported that these sequences can directly detect signals from non-aqueous myelin protons in the white matter of the brain [16,17,20,42,43,44,45]. Horch et al. investigated the origins of the ultrashort T2 proton NMR signals in myelinated nerves. They suggested that UTE sequences could be used to directly measure the ultrashort T2 signals (50 µs < T2 < 1 ms) as a new means of quantitative myelin mapping [16]. Wilhelm et al. examined UTE imaging of purified bovine myelin extract and rat thoracic spinal cord samples on a 9.4 T spectrometer (Bruker DMX 400). They found myelin T2*s varied between 8 µs and 26 ms with ~90% of the myelin T2* less than 1 ms [17]. Weiger et al. investigated myelin imaging with a 3D ZTE-based technique with hybrid filling (HYFI) on a 3T whole-body scanner and found 85% of the myelin signal had a T2* of 7.5 µs [20]. Sheth et al. studied UTE imaging of lyophilized bovine myelin powders and reported a T2* of 110–160 µs [43]. Ma et al. reported a novel 3D short TR adiabatic inversion recovery UTE (STAIR-UTE) technique for myelin mapping in vivo and reported a short T2* of ~210 µs [44]. Shen et al. applied dual-echo UTE with a rosette k-space pattern to the brain and reported a T2* of ~0.10 ± 0.06 ms and a fraction of 10.9% ± 1.9% for myelin in white matter and a T2* of 0.09 ± 0.12 ms and a fraction of 5.7% ± 2.4% for myelin in white matter [46]. The considerable variations in myelin T2* and fraction values suggest that more studies are needed to explore the signal sources in UTE and ZTE imaging of myelin. More studies are also necessary to explain why collagen backbone protons are invisible but myelin protons are visible. It is likely related to the increased mobility of non-aqueous protons in myelin over the more rigid collagen backbone protons in bone and tendons.

There are several limitations in this study. First, it is unclear whether the D_2_O–H_2_O exchange and subsequent freeze-drying procedures would affect the collagen structure. Freeze-drying may result in slight protein denaturation and the disruption of water bridges, which are involved in stabilizing the structure [47]. However, the changes in collagen structure, if any, are likely small [23] and are unlikely to significantly affect the transverse relaxation times of collagen backbone protons. We expect the conclusion that ZTE cannot detect signals from collagen backbone protons still held in fresh bone, tendons, and other collagen-rich connective tissues in the MSK system. Second, repeated D_2_O–H_2_O exchange and freeze-drying may lead to significant tissue degradation. Again, we expect that this degradation should not significantly affect the transverse relaxation times of collagen backbone protons. Therefore, the conclusion that collagen backbone protons are invisible with the ZTE sequence is still valid. Third, the D_2_O–H_2_O exchange process removes all exchangeable protons, leaving unexchangeable protons in the collagen structure (e.g., -CH2-, -CH3) [48]. The pure noise images in ZTE imaging of bone and tendon samples post D_2_O–H_2_O exchange and freeze drying could only demonstrate that the unexchangeable protons were invisible. There is no direct evidence that exchangeable protons in the collagen structure are invisible with the ZTE sequence. Fourth, the longitudinal and transverse relaxation times of collagen backbone protons are unknown. As far as we know, there are no papers reporting T2 or T2* for collagen backbone protons in bone and tendons. The T2 of ~10 µs derived from MT modeling is indirect and might be inaccurate. It is difficult to explain why collagen backbone protons with T2* ~10 µs are invisible with the ZTE sequence, but non-aqueous myelin protons with T2* ~10 µs or less are visible with the ZTE sequence. Further research is necessary to enhance our understanding of this interesting topic. Finally, our preliminary results demonstrate that collagen backbone protons are invisible with UTE and ZTE sequences. We expect collagen backbone protons in bone and tendons to be invisible with other UTE-type sequences, including WASPI, SWIFT, AWSOS, and PETRA sequences.

## 5. Conclusions

The strong ZTE signal from fully hydrated bone and tendon samples but pure noise after D_2_O exchange and freeze-drying suggest that collagen backbone protons are “invisible” with the ZTE sequence.

## Figures and Tables

**Figure 1 bioengineering-12-00016-f001:**
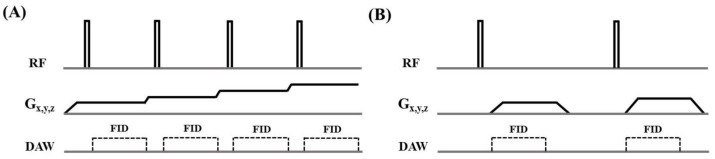
The 3D ZTE sequence utilizes a short rectangular RF pulse (duration = 8 µs, flip angle = 4°) for nonselective excitation, followed by 3D center-out radial sampling during fully ramped-up readout gradients (**A**). The 3D UTE sequence employs a short rectangular RF pulse (duration = 32 µs, flip angle = 10°) for nonselective excitation, followed by 3D center-out radial ramp sampling (**B**).

**Figure 2 bioengineering-12-00016-f002:**
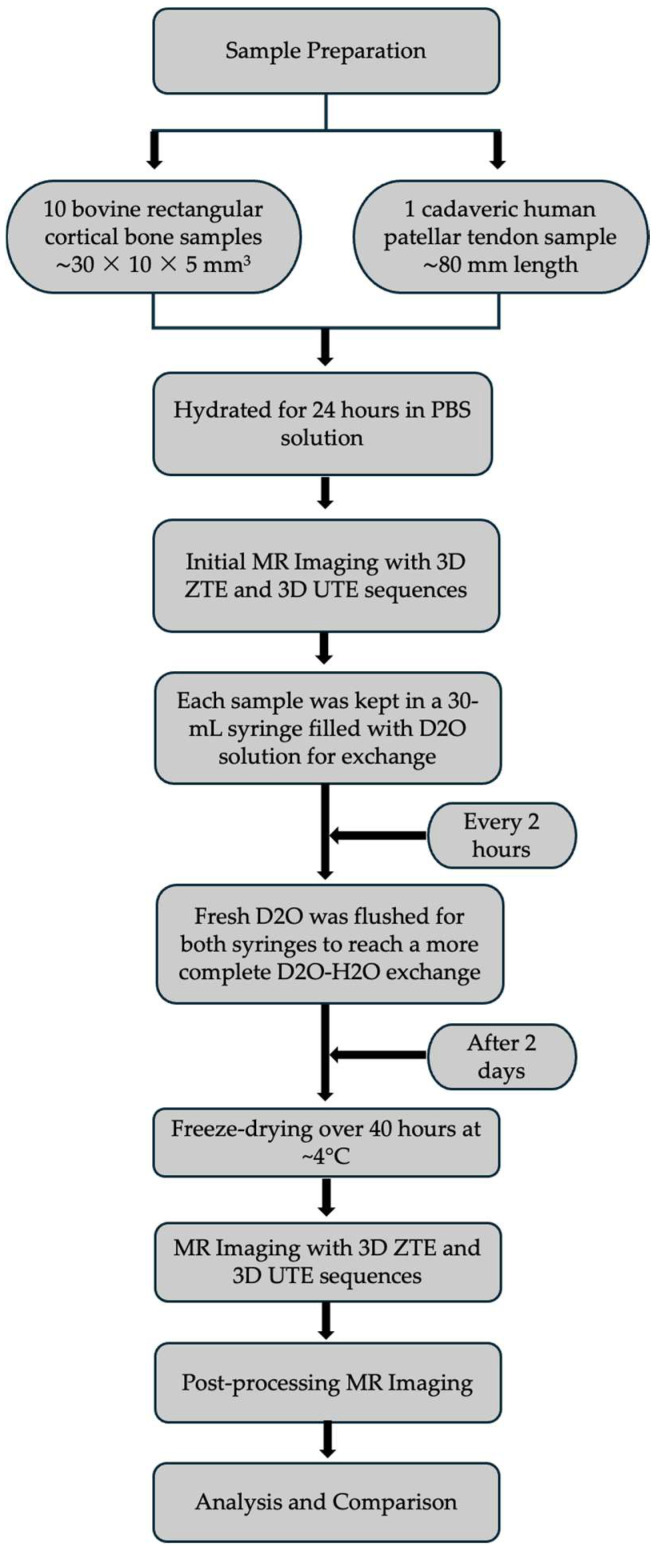
A flow diagram for the experimental procedure. PBS, phosphate-buffered saline.

**Figure 3 bioengineering-12-00016-f003:**
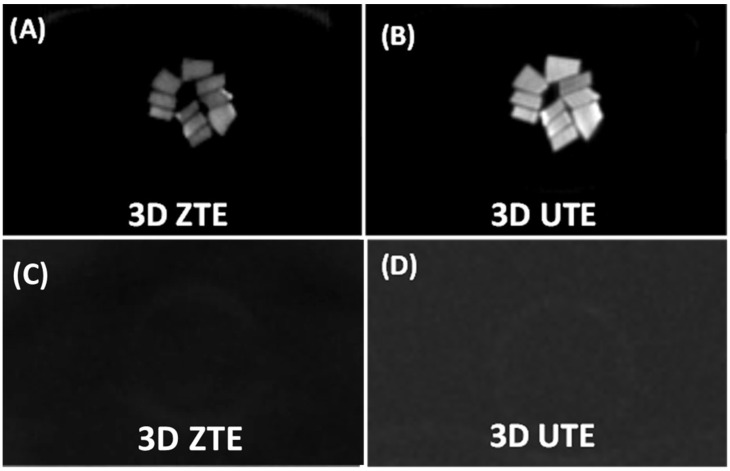
Fully hydrated bovine cortical bone samples were imaged with 3D ZTE (**A**) and UTE sequences (**B**), along with ZTE (**C**) and UTE (**D**) imaging of the same bone specimens after two days of repeated D_2_O exchange followed by freeze-drying for over 40 h. Both the ZTE and the UTE sequences show high signals for the hydrated cortical bone samples but zero signals after the repeated D_2_O exchange and freeze-drying procedure.

**Figure 4 bioengineering-12-00016-f004:**
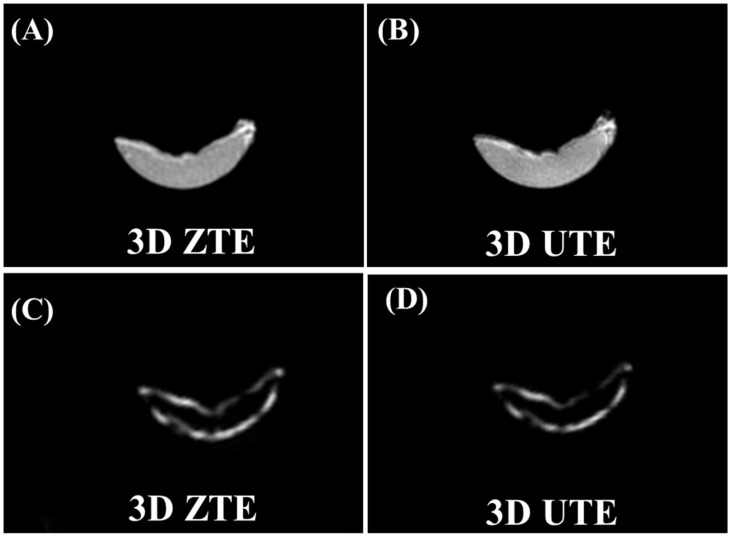
A fully hydrated cadaveric human patellar tendon sample was imaged with 3D ZTE (**A**) and UTE sequences (**B**), along with ZTE (**C**) and UTE (**D**) imaging of the same patellar tendon specimen after 2 days of repeated D_2_O exchange followed by freeze-drying for over 40 h. The hydrated patellar tendon sample shows a high signal with both ZTE and UTE sequences. After the repeated D_2_O exchange and freeze-drying procedure, only thin bright lines were observed, which showed typical fat/water in-phase and out-phase behaviors.

**Table 1 bioengineering-12-00016-t001:** MRI parameters of the 3D ZTE and UTE sequences.

Parameters	ZTE Sequence	UTE Sequence
Repetition Time (TR)	2.1 ms	10 ms
Echo Time (TE)	12 µs	28 µs
Flip Angle (FA)	4°	10°
Pulse Duration	8 µs	32 µs
Receiver Bandwidth	62.5 kHz	
Field of View (FOV)	40 mm	
Acquisition Matrix	Bone: 192 × 192 × 16 Tendon: 256 × 256 × 16	
Slice Thickness	3 mm	
Scan Time	2 minutes	

**Table 2 bioengineering-12-00016-t002:** Summarize the SNR and CNR values of the 3D ZTE and UTE images.

MRI Sequences	Tissue	Condition	SNR (Mean ± SD)	CNR (Mean ± SD)
ZTE	Bone	Hydrated	54.6 ± 3.1	46.2 ± 3.1
		D2O Exchange + Freeze-Dried	5.36 ± 1.11	0.03 ± 1.11
	Tendon	Hydrated	77.0 ± 7.3	73.1 ± 7.0
		D2O Exchange + Freeze-Dried	8.53 ± 2.18	−0.09 ± 0.58
UTE	Bone	Hydrated	103.1 ± 9.8	98.4 ± 9.8
		D2O Exchange + Freeze-Dried	5.64 ± 0.08	0.015 ± 0.082
	Tendon	Hydrated	108.3 ± 4.4	102.6 ± 4.2
		D2O Exchange + Freeze-Dried	4.95 ± 0.60	−0.7 ± 1.6

## Data Availability

The original contributions presented in this study are included in the article. Further inquiries can be directed to the corresponding author(s).

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
