# Peer review of "Are Collagen Protons Visible with the Zero Echo Time (ZTE) Magnetic Resonance Imaging Sequence: A D2O Exchange and Freeze-Drying Study"

_bioengineering, 2024, doi:10.3390/bioengineering12010016_

Round 1

Reviewer 1 Report

Comments and Suggestions for Authors

This work by Guo et al. reports an interesting experiment using UTE/ZTE MRI to probe collagen protons. The authors used D2O, which does not give MR signals, to gradually replace protons (H) in tissues. The hypothesis is that exchangable protons will be replaced by D2O and collagen protons, or at least the backbone protons, will remain and be detectable by MRI. They used UTE/ZTE due to potential low T2/T2* of collagen protons. After 2 days preparation, the authors did not observe any signals. This is an interesting finding, and the authors listed two potential causes: (1) tissue degradation causing all collagen protons to be replaced by D2O; (2) collagen protons have extremely short T2/T2* (< 10 us).

Overall, the study was performed well but I wonder whether additional evidence can be provided. For example, can spectroscopy techniques from solid state MRI be used to evaluate the collagen proton. Will data from intermediate steps (e.g. 1 day after D2O exchange) provide additional insights?

References on how MT was used to determine the T2 of collagen proton should be provided. 

Author Response

This work by Guo et al. reports an interesting experiment using UTE/ZTE MRI to probe collagen protons. The authors used D2O, which does not give MR signals, to gradually replace protons (H) in tissues. The hypothesis is that exchangeable protons will be replaced by D2O and collagen protons, or at least the backbone protons, will remain and be detectable by MRI. They used UTE/ZTE due to the potential low T2/T2* of collagen protons. After 2 days of preparation, the authors did not observe any signals. This is an interesting finding, and the authors listed two potential causes: (1) tissue degradation causing all collagen protons to be replaced by D2O; (2) collagen protons have extremely short T2/T2* (< 10 us).

Overall, the study was performed well, but I wonder whether additional evidence can be provided. For example, can spectroscopy techniques from solid-state MRI be used to evaluate the collagen proton? Will data from intermediate steps (e.g., 1 day after D2O exchange) provide additional insights?

References on how MT was used to determine the T2 of collagen proton should be provided. 

Response: We thank the reviewer for the excellent comments and suggestions. We agree with the reviewer that spectroscopy techniques from solid-state MRI would be helpful in evaluating the collagen protons, including their T1 and T2 relaxation times. Unfortunately, we don’t have access to solid-state MR spectroscopy.

Data from intermediate steps might provide information on how fast D2O and H2O exchange in cortical bone and tendons. A few studies have demonstrated that exchange rates are related to collagen cross-linking. Please refer to the papers below:

  1. Fishbein KW, Gluzband YA, Kaku M, et al., Effects of formalin fixation and collagen cross-linking on T2 and magnetization transfer in bovine nasal cartilage. Magn Reson Med. 2007; 57:1000-1011.
  2. Ho LC, Sigal IA, Jan NJ, et al. Non-invasive MRI Assessments of Tissue Microstructures and Macromolecules in the Eye upon Biomechanical or Biochemical Modulation. Sci Rep. 2016;6(1):32080.
  3. Gochberg DF, Fong PM, Gore JC. Studies of magnetization transfer and relaxation in irradiated polymer gels - interpretation of MRI-based dosimetry. Phys Med Biol. 2001;46(3):799.

A systematic study on exchange rates between free/bound water and collagen backbone protons is important and may be very helpful in explaining the mechanical degradation of bone/tendon. For example,  T2D is characterized by normal or high bone mineral density (BMD) but 40~70% higher hip fracture risk. One potential explanation is related to high glucose levels, which lead to the creation of advanced glycation end-products (AGEs), causing non-enzymatic crosslinking, thereby increasing the brittleness of the otherwise elastic collagen fibers and reducing bone toughness. Collagen may have less of an effect on bone strength and stiffness than mineral; it has a profound effect on bone fragility. It is the primary toughening mechanism in bone.

This study aims to investigate whether ZTE and UTE MRI sequences can detect collagen backbone protons in cortical bone and tendons using a whole-body clinical scanner. Information from this intermediate step should not affect the conclusion of our study.

We have added references on how MT was used to determine the T2 of collagen protons.

Reviewer 2 Report

Comments and Suggestions for Authors

This study presents an investigation into the potential of ZTE MRI sequence on a clinical 3T scanner to image collagen protons directly via D2O exchange and freeze-drying experiments. Although the paper is written well, the novelty of the proposed work is not clear. Moreover, limited experimental results are provided in this study in support of the investigation conducted here. The following suggestions are recommended to the authors to address the above issues and improve the paper as well.

Comments

  1.   The scientific significance of detecting the changes in water and collagen in the musculoskeletal (MSK) tissues should be more elaborated in the first paragraph of the Introduction.

  2.   Several subjects-specific terms such as ‘transverse relaxation time’, ‘tissue T2*s, ‘transverse magnetization’, ‘radial ramp sampling’, ‘k-space center’, ‘D2O exchange’, and so on, should be briefly defined to make the problem investigated in this study more comprehensible to the general readers. Moreover, the procedure of spatial encoding and its uses for the problem under investigation in this study should also be briefly discussed in the 2nd paragraph of the Introduction.

  3.   The structure of collagen proton along with UTE and ZTE sequences should be illustrated to clarify the specific issue stated in the third paragraph of the Introduction in a better way.

  4.   The contributions of the proposed study should be pointed out in the listed form at the end of the Introduction.    

  5.   All abbreviated terms must be defined in a separate table at the beginning of Section 3.

  6.   Ten bovine cortical bone samples sectioned in a rectangular shape and the cadaveric human patellar tendon sample should be pictorially shown in Section II for better visualization.

  7.   The instrument used for, and various parameters involved in the 3D ZTE and UTE imaging process along with their values should be displayed in a table.

  8.   A flow diagram for the experimental procedure should be provided for its better understanding.

  9.   The results for both the UTE and ZTE imaging in terms of SNR and CNR values under two different experimentation scenarios should be shown in tabular form.

  10.   More experimentation using different values of the parameters like flip angle, excitation pulse, etc., for both UTE and ZTE imaging should be carried out to demonstrate the effect of formers on the performances of the latter.  Experimental results in terms of other metrics like T2*, gravimetric analysis, amino acid analysis, etc., should also be provided to compare the performances of the proposed procedure with the existing techniques.

Round 2

Reviewer 2 Report

Comments and Suggestions for Authors

The authors have addressed all my concerns well by providing a point-to-point reply to each raised concern and also updated the manuscript accordingly.